# Evaluation of Four Thermal Comfort Indices and Their Relationship with Physiological Variables in Feedlot Cattle

**DOI:** 10.3390/ani13071169

**Published:** 2023-03-25

**Authors:** Rodrigo A. Arias, Terry L. Mader

**Affiliations:** 1Instituto de Producción Animal, Facultad de Ciencias Agrarias y Alimentarias, Universidad Austral de Chile, Valdivia 5090000, Chile; 2Centro de Investigación de Suelos Volcánicos, Universidad Austral de Chile, Valdivia 5090000, Chile; 3Animal Science Department, University of Nebraska-Lincoln, Lincoln, NE 68583-0908, USA

**Keywords:** heat stress, tympanic temperature, respiration rate

## Abstract

**Simple Summary:**

Heat waves have been a more recurrent phenomenon in the last decade, which in turn creates a challenging environment for feedlot cattle. Producers require tools that are easy to implement in order to adopt adequate mitigation techniques to minimize the adverse effects of heat waves on animal wellbeing and animal performance. This study assesses four different thermal comfort indices in order to predict the tympanic temperature of feedlot steers. The temperature humidity index adjusted by solar radiation and wind speed (THI_adj_) as well as THI estimated using pen surface temperature (THI_PST_) were demonstrated to be practical tools for predicting animal wellbeing.

**Abstract:**

Climatic data from different years and experiments conducted in Nebraska were used to estimate four comfort thermal indices and to predict the risk of heat stress and its relationship with pen surface temperature (PST). These included the temperature–humidity index (THI), the adjusted THI (THI_adj_), the heat load index (HLI), and THI_PST_ using pen surface temperature instead of air temperature. Respiration rates (RR), tympanic temperatures (TT), and panting scores (PS) were also collected in each year and from each location. During 2007, mean values of soil temperature, PST, outgoing shortwave radiation, and TT were greater than in 2008 (*p* < 0.011). However, HLI, relative humidity, and incoming and outgoing long-wave radiation were greater during 2008 (*p* < 0.012). The TT was positively correlated with THI_PST_ and THI_adj_ (0.75 and 0.70, respectively), whereas RR had a moderate correlation with THI, THI_adj_, and HLI (0.32, 0.27, and 0.34, respectively; *p* < 0.001). Thermal comfort indices showed a positive correlation with TT, especially the THI_PST_. These relationships vary with location. However, all of the thermal indices showed weak relationships with the observed RR. This would confirm the different roles that TT and RR have as indicators of heat stress. The THI_PST_ was the best index for predicting TT across years.

## 1. Introduction

The microclimate surrounding cattle exerts a powerful effect on animals’ health, productivity level, and survival [1]. Thus, any change in one or more of these variables may impact animal well-being and productivity. To date, the effects of ambient air temperature (Ta) have been widely studied on cattle, whether alone or in conjunction with relative humidity [2,3,4]. Proportionally, much less attention has been given to other environmental factors such as wind speed (WS) and solar radiation (SR). Nevertheless, it has been recognized that SR and WS affect animal energy balance by modifying the total heat load of cattle [5]. Adjustment for SR has been made to the temperature humidity index (THI), a popular thermal comfort index [6,7]. In addition, SR has been also used to estimate the respiration rate index [8,9] and to predict heat stress in cattle. Furthermore, it has been reported that SR increases rectal temperature and reduces fertility in dairy cattle [10].

The uncomfortable microclimate for cattle is exacerbated in the summer season by the increasing and recurrent heat waves, which impact the well-being, performance, and survival of feedlot cattle around the world [1,11]. The excessive heat load to which cattle are exposed during these periods demands a practical tool that allows for the quantification of thermal stress under commercial conditions, because it causes significant losses. In fact, heat stress has become an important issue for livestock industries [11,12,13]. For almost 40 years, the temperature–humidity index (THI), a single value representing the combined effects of air temperature and humidity [14], has been used as a tool for thermal stress risk in humans and livestock. St-Pierre et al., [12] estimated that 43% of the total economic losses in the USA were due to the heat stress of all livestock occurring in five states (Texas, California, Oklahoma, Nebraska, and North Carolina). Consequently, the development of practical tools that allow for the adequate handling of cattle under these unfavorable environmental conditions has become a major concern [15,16,17].

Heat stress has been defined as a multifactorial problem limiting animal efficiency and negatively impacting animal health [18]. Mellor [19] and Webster [20] indicate that animal husbandry requires more than just supplying the basic needs of animals (water, food, shelter, and shade), making particular mention of meteorological conditions to which cattle are exposed. In fact, the five domains model recently proposed has a predominantly physiological orientation to assess “physical/functional disruptions and imbalances, as well as restrictions on behavioral expression, and then to identify the specific negative affects each disruption, imbalance, or restriction would be likely to generate” [19]. These domains include “nutrition”, “environment”, “health” and “behavior”, and a fifth, “mental”, that together have an accumulated impact on animal welfare. In this manuscript, we present measurements of both environmental (e.g., SR and WS) as well as animal-based animal welfare indicators (e.g., RR and PS) contributing to a better understanding from this standpoint.

In 1970, the THI was adopted by the livestock industry in the USA as a basis for the Livestock Weather Safety Index [21], which describes four categories of heat stress risk and has been used as a heat stress indicator around the world [17]. However, the THI has been widely criticized, since it does not account for solar radiation and wind speed, two critical factors that affect the thermal balance in cattle [7,22], or genotype differences [17,23]. In addition, Collier and Zimbelman [24] assert that THI underestimates the effects of heat stress on cattle. Due of this, adjustments to the THI have been proposed, including a combination of wet- and dry-bulb temperature [25] as well as adjustments for solar radiation and wind speed [7]. Thus, this paper aimed to assess the relationship among four thermal comfort indices used to predict the risk of heat stress and their relationship with the pen surface temperature at feed yards.

## 2. Materials and Methods

Experiments carried out in 2007 in Concord were conducted at the University of Nebraska Haskell Agricultural Laboratory, previously reported by Mader et al. [26], located at 42°23′ N latitude and 96°57′ W longitude; elevation 445 m. For this study, pens with no shelter or windbreak to the north, west, and northwest were used. Similarly, pens at the Panhandle Research and Extension Center (PHREC) feedlot at Scottsbluff, Nebraska (41°51′ N, 103°40′ W; elevation 1186 m) also corresponded to open pens with no windbreaks and no shelter/shade (2008).

### 2.1. Micrometeorological Data Collection and Thermal Indices Estimation

All of the data correspond to three experiments conducted during the summer season in two experimental feedlots of the University of Nebraska (2007 and 2008) located in Concord and Scottsbluff, NE, USA, and reported by Arias [15]. Meteorological data (ambient temperature (Ta), black globe temperature (BG), relative humidity (RH), wind speed (WS), solar radiation (SR), precipitation, soil temperature at 10 cm depth, and pen surface temperature) were collected using weather stations located at the feedlots. In Concord’s experiments (2007 and 2008), environmental data were collected continuously in a weather station data logger CR10X (Campbell Scientific Inc., North Logan, UT, USA) and summarized by the time of day. The weather station was located at the center of the fence line dividing the two central pens of the alley. Soil temperature (10.2 cm depth) was recorded using a thermistor model 107 (Campbell Scientific Inc., North Logan, UT, USA), which was located within a tube buried next to the fence that separates the pens and attached to the weather station. Pen surface temperature was recorded using a laser infrared gun located approximately 2.0 m above ground. The laser gun was attached to the weather station and directed to the center of the mound in the center of the pen (approximately 1.5 m from the fence) an area commonly used by the animals. Wind speed was recorded using an anemometer model 014A (Met One Instruments, Inc., Grants Pass, OR, USA), whereas air temperature and relative humidity were recorded with an HMP35 sensor (Campbell Scientific Inc., North Logan, UT, USA). Net solar radiation was obtained from the High Plains Climate Center automated weather station located 0.6 km west and 1.5 km north of the feedlot near Concord, NE. On the other hand, in the Scottsbluff experiment (2008), pen surface temperature was collected with a laser gun (IRtec MicroRay Xtreme, E instruments Group LLC, Langhorne, PA, USA) mounted in a tripod and directed to the center of the pen. The soil temperature was recorded using an iButton buried at a depth of 10.2 cm (DS1922L, Nexsens Technology, Beavercreek, OH) next to the fence that divides two pens. These data were used to estimate four indices of thermal comfort for cattle, including the THI, adjusted THI (THI_adj_), and heat load index (HLI), which were calculated using the following equations:

(a) Temperature–humidity index [2]:THI = 0.8 × Ta + [(RH/100) × (Ta − 14.4)] + 46.4
where Ta is the air temperature (°C) and RH is the relative humidity (%);

(b) Adjusted temperature–humidity index [6,27]:THI_adj_ = THI + 4.51 − (1.992 × WS) + (0.0068 × SWin)
where THI is the temperature humidity index, WS is the wind speed (m·s^−1^), and SWin is the incoming shortwave radiation (W·m^−2^);

(c) Heat load index [17]:HLI_(BG>25)_ = 8.82 + (0.38 × RH) + (1.55 × BG) − (0.5 × WS) + e^(2.4−WS)^
HLI_(BG<25)_ = 10.66 + (0.28 × RH) + (1.3 × BG) − WS
where BG is the black globe temperature [25] and e is the base of the natural logarithm.

The black globe temperature was estimated for Scottsbluff by using the following equation [23]:BG = (1.33 × Ta) − (2.65 × (Ta)^(1/2)^) + (3.21 × log(SWin + 1)) + 3.5

(d) Estimated respiration rate [9]:RRe = (5.1 × Ta) + (0.58 × RH) − (1.7 × WS) + (0.039 × SR) − 105.7

The black globe temperature was used to estimate HLI in Scottsbluff [28].

In addition, a THI index based on pen surface temperature (THI_PST_) was also estimated. The THI_PST_ was calculated using pen surface temperature instead of air temperature. Additionally, respiration rates (RR), TT, and panting scores (PS) were also collected in each year and location. Finally, net solar radiation (NR), incoming shortwave solar radiation (SWin), outgoing shortwave solar radiation (SWout), incoming long-wave radiation (LWin), and outgoing long-wave radiation (LWout) were collected by two precisions spectral pyranometers (Eppley Lab., Inc., Newport, RI, USA), whereas incoming and outgoing long-wave radiation were collected using two precision infrared radiometers (Eppley Lab., Inc., Newport, RI, USA). Simultaneously, net solar radiation was collected using an REBS Net Radiometer model Q − 7.1 (Radiation and Energy Balance Systems, Inc., Seattle, WA, USA). All radiation measurements were collected hourly in an adjacent empty pen in the feed yard.

### 2.2. Physiological Data Collection

Experiments 1 and 2 (Concord, NE 2007 and 2008): in the first experiment, 34 crossbred steers (Angus × Hereford) were fitted with an iButton^®^ logger in order to collect TT for a period of six days (11 to 17 July). In experiment 2, a total of 35 steers received the same kind of loggers (14 to 19 July). Steers receiving the devices were chosen randomly each year. The iButtons loggers were programmed to collect TT once every minute and then compiled it into hourly readings. The devices were inserted manually into the ear canal of each animal as described by Arias, et al. [29]. Additionally, RR and PS were recorded between 14:30 and 15:30 h each day by visual observation of the animals.

A score of PS was assigned using the scoring system presented in Table 1. The RR was determined by counting 10 flank movements (timing of 10 breaths) of each steer with a stopwatch. Experiment 3 (Scottsbluff NE, 2008): in this experiment, ten steers were fitted with iButton^®^ loggers between 22 and 25 July 2008. Data from TT, RR, and PS were collected following the same procedures previously mentioned.

### 2.3. Statistical Analysis

Analyses of the relationships between thermal comfort indices and physiological stress indices were investigated. These analyses were conducted using JMP^®^ and SAS^®^, whereas Excel for Mac^®^ was used for plotting mean values. Additionally, comparisons between years and locations were conducted for the thermal indices and the main environmental variables by ANOVA. The relationship between the observed and calculated respiration rates was also investigated, as well as the relationship between PS and observed and estimated respiration rates. These relationships were conducted by using a subset of data that contained the average value of the environmental and physiological variables collected between 12:00 and 15:00 h, because this was the time at which PS and RR were collected. The final analyses included correlation (PROC CORR and PROC MEANS in SAS) and regression analysis (JMP). The experimental and observational unit in each analysis was a steer and the significance level was established at 5%.

## 3. Results

### Climate Conditions

Table 2 summarizes the climatic conditions for the whole data set by year. No differences between years were observed for WS, Ta, THI, THI_adj_, THI_PST_, and SWin. During 2007, the mean values of soil temperature, pen surface temperature, SWout, and TT were greater compared to those measured in 2008 (*p* < 0.011). In comparison, the mean values for HLI, RH, LWin, and LWout were greater during 2008 (*p* < 0.012). In 2008 there was a greater variation in the climatic variables. For instance, RH had a range of 72.0 vs. 59.3% for the years 2008 and 2007, respectively. Soil temperature and PST had a range of 12.3 vs. 9.5 °C and 57 vs. 36.5 °C, respectively, whereas TT and the Ta had a range of 3.2 vs. 2.1 °C and 19.5 vs. 24.3 °C for years 2008 and 2007, respectively. There were differences in the soil temperature, RH, THI, THI_adj_, SWout, LWin, LWout, and HLI (*p* < 0.015) between locations, with greater values at Concord than at Scottsbluff. In addition, Ta tended to be greater in Concord (*p* = 0.07), whereas pen surface temperature tended to be higher in Scottsbluff (*p* = 0.07), which was a sandier soil.

There was more variation in RH in Scottsbluff, which may be due to the precipitations recorded during the week of data collection. Correlation coefficients for all of the thermal comfort indices, TT, and RRe are presented in Table 3. Tympanic temperature was highly and positively correlated with all of the thermal indices. However, the greatest values were observed for THI_PST_ and THI_adj_ (0.75 and 0.70, respectively).

Similarly, RRe was correlated positively with all of the thermal indices, but mainly with THI_adj_, THI, and THI_PST_ (0.96, 0.93, and 0.85, respectively), whereas correlation with TT and HLI was lower (0.68 and 0.59, respectively).

Table 4 and Table 5 show the coefficients of correlation for the variables by location. In Concord, TT was highly correlated with RRe, THI_PST_, and THI_adj_ (0.94, 0.80, and 0.70, respectively), whereas a weak relationship was observed with HLI (0.22). However, HLI showed a better correlation coefficient with RRe. A simple linear regression analysis showed that THI explained 44 and 87% of the variability of TT and RRe, respectively. Likewise, when THI_adj_ was used as a predictor the results improve lightly (46 and 92% of the TT and RRe, respectively). However, the best predictor for TT was obtained with THI_PST_ (53%), which in turn explained 72% of the variation in RRe. Finally, HLI showed lower values with 0.38 and 0.10 for RRe and TT, respectively. In general, TT was explained better by all thermal indices in Concord than in Scottsbluff, except for HLI. In addition, RRe explained more variability of TT in Scottsbluff than in Concord. Interestingly, in both locations, the best variable explaining TT was the THI_PST_. A similar response was observed when data from both locations were pooled. On the other hand, the THI and THI_adj_ were the best thermal comfort indices that explained the RRe in Concord, whereas the THI_PST_ and HLI were the best ones in Scottsbluff.

Figure 1 shows the relationships between the thermal comfort indices and the physiological indices (TT and RRe). In general, HLI, THI, and THI_adj_ showed similar patterns during a 24 h period, with HLI reaching the maximum value early in the morning (09:00 h). The HLI remained very constant until 17:00 h, after that, it started to decrease quickly. In addition, THI and THI_adj_ showed similar patterns but with slightly higher values for THI_adj_ during the period of the day with higher solar radiation, that is, between 10:00 and 17:00 h. This highlights the importance of SWin in THI_adj_ estimation. It is important to mention that there was a lag of maximum TT regarding the maximum values of all comfort thermal indices. This lag was greater in Concord than in Scottsbluff (4 vs. 1 h).

The accuracy of the equation used to calculate RRe was assessed by comparing these data with collected respiration rates (RR) during the summer of 2008. A low correlation was observed for RRe and RR (0.27; *p* < 0.0001). Figure 2 shows that the observed RR was greater than those estimated by Eigenberg’s equation (111.75 ± 0.82 vs. 104.37 ± 0.76, respectively; *p* < 0.0001) when data from both locations and years were analyzed (Figure 2a). Similarly, when only data from Scottsbluff were analyzed (Figure 2b), they also presented a greater value in the observed RR (122.19 ± 2.45 vs. 103.42 ± 2.45, *p* < 0.0001).

Finally, the relationships between respiration rates (RR and RRe) and PS are presented in Figure 3. A simple linear regression analysis shows that PS explained 82% of the observed RR, whereas RRe only explained 5%.

## 4. Discussion

Thermal comfort and heat stress indices have been the subject of in-depth studies in man and livestock [5,30]. However, the assessment of thermal stress in terms of psychological responses is complex [30]. In livestock, the primary focus has been on indices that support rational environmental management decisions related to performance, health, and well-being [5,31]. Thus, during the last twenty years, the THI has become a de facto standard for classifying thermal environments in many animal studies [5]. However, this index has been the subject of diverse criticism by diverse researchers, because many times, it overestimates or underestimates the real effects of heat stress in cattle [17,24].

No differences were observed between years for THI and THI_adj_. This could be explained in part because Ta was similar between years, even though there were higher RH values during 2008, which apparently did not impact the THI indices. In addition, no differences were observed in WS and SWin between years, which are the factors used by Mader et al. [7] to adjust THI. Likewise, the higher value of THI_PST_ in 2007 was due to the higher values recorded for soil surface temperature for that year. On the other hand, HLI was greater during 2008 because of higher RH and black-globe temperatures (27.2 ± 0.8 vs. 29.4 ± 0.7 for 2007 and 2008, respectively). Thus, none of the thermal comfort indices assessed were able to match the difference observed for TT between years. Only THI_PST_ showed a better result that could explain the differences in TT observed. Based on the Livestock Weather Safety Index [21], the greater value for THI_PST_ (77.7) corresponded to a condition of “alert”, whereas all of the other thermal indices were considered “normal”.

When data were compared across locations, no differences were observed in TT, even though THI, THI_adj_, THI_PST_, and HLI were statistically greater in Concord than in Scottsbluff. Nevertheless, the indices presented normal values (<74), and HLI was considered safe (<86). The greater values of THI and THI_adj_ could be associated with the higher values of RH recorded in Concord. However, an analysis of the number of hours of thermal indices above “danger” for the THI (>79) and “extreme caution” for HLI (>92) showed that, in 2007, there were more hours under these categories. Simple correlation analyses for the number of hours of thermal indices over critical values were positively correlated with TT in 2007 (0.87, 0.92, 099, and 0.39 for THI, THI_PST_, THI_adj_, and HLI, respectively). These relationships were less clear during 2008 due to fewer hours over those critical values. Only the THI_PST_ showed a consistent relationship trend for Concord and Scottsbluff during 2008 (0.39 and 0.60, respectively). For the experimental periods, THI_PST_ presented a greater number of hours over the “danger” category in 2007 (Concord) and 2008 (Scottsbluff), but in 2008 (Concord), THI_adj_ was the index with a greater number of hours in that category. It is important to recall that the five domains’ concept implies not only providing the basics to animal survival but also reaching a condition of well-being. Thus, tools which are easy to adopt and use, like those here proposed, are of high relevance to producers for their animals to reach a welfare enhancement status.

The high correlations observed between RRe and the thermal comfort indices could be due to the equations used to estimate RR and THI sharing at least two of the same environmental variables (RH and Ta). The use of Eigenberg’s equation (RRe) to obtain respiration rates in this analysis could be confounding the real relationship of the comfort indices with the respiration rates of cattle, because most of them are based on the same environmental variables.

Therefore, only the relationships of TT with the indices and the estimated RR should be considered. The same observation is valid for the linear regression analysis herein conducted for the RRe. Assessing the relationships of RRe and RR with TT, a new set of correlation coefficients for RR and the variables of interest were calculated. These correlations were conducted using the observed RR collected during the afternoon. The results showed a moderate correlation with THI, THI_adj_, and HLI (0.32, 0.27, and 0.34, respectively; *p* < 0.0001). Likewise, the correlation coefficients between PS and the thermal indices were 0.29, 0.19, and 0.25 for THI, THI_adj_, and HLI, respectively (*p* < 0.0001). The correlation coefficients between TT and PS and between TT and RR were lower (0.29 and 0.36, respectively; *p* < 0.0001). However, the correlation coefficient between RR and PS was high (0.90; *p* < 0.0001). Non-significant correlations were observed for THI_PST_ with PS or RR, but this was significant with tympanic temperature (0.19; *p* < 0.0001)

In a study conducted by da Silva et al. [28] assessing the efficacy of six different thermal indices under tropical conditions in Brazil, THI and the black globe-humidity index had the lowest correlations with rectal temperature and RR in data recorded in dairy farms (Holstein cows). On the contrary, results herein presented indicate higher correlation coefficients for THI and THI_adj_ with TT, but similar values of correlation for HLI. In their experiments, da Silva et al. [28] did not assess the THI_adj_ index. The same authors reported that the best correlation coefficients among thermal indices and RR were obtained with HLI and the equivalent temperature index [32] were 0.54 and 0.52, respectively. Our study observed lower correlation coefficients with a maximum of 0.34 and 0.32 for HLI and THI, respectively. On the other hand, Gaughan et al. [17] reported similar values of r^2^ for TT and THI (0.26) to those reported by da Silva et al. [28]. However, the results herein presented showed greater r^2^ for THI, THI_adj_, and THI_PST_ (0.44, 0.46, and 0.53, respectively). Differences among the studies discussed can have many potential explanations, e.g., micrometeorological conditions in each location due to relief and microrelief, wind exposition, and soil texture (sandy soil in Scottsbluff). Changes in soil density imply changes in the proportion of porous space and associated changes in the water retention of a particular soil. Both are important factors related to the heat flow capacities of a particular soil and its potential transfer to animals. Nevertheless, no reports regarding the relationship between pen surface temperature and animal production or animal physiology have been published up to now. Nevertheless, the exchange of heat at the ground surface exposed to sunshine establishes the conditions of an important part of the animal’s environment [33]. In addition, there is high individual variation among animals, including metabolism, tissue composition, behavior, as well as adaptation, in addition to animal management, diet composition, facilities, water intake, and water quality, among many other factors.

Body temperature and RR lags have been previously reported, ranging from 1 to 4 h [8,34]. Even though there is a positive relationship between RR and TT, there was a great difference in how thermal indices related to them. These differences could be explained by the role each of these variables plays in the animal heat balance. The RR is a mode of thermo-regulation, while the TT is the result of thermal equilibrium [35]. Therefore, there is an inconsistency in the result observed in this report and previous investigations regarding the relationship between thermal indices and TT. These differences reveal the complexity of obtaining a single index that attempts to accurately estimate the risk of heat stress under commercial conditions, because, as mentioned previously heat stress is a multifactorial problem. Consequently, there is a response that is highly variable among individuals. For example, Brown–Brandl [36] reported that two heifers under the same weather conditions, in the same feedlot, and under the same management presented a higher RR of 78 and 167 bpm (at 32.9 °C). However, even with those differences, the RR is only a picture of the specific moment of the day and does not necessarily represent the thermal status of an animal. Another difficulty lies in the estimation of the best predictor of the environment to which the animal is exposed during the period (usually a 24 h period), given that climatic variables follow a circadian pattern, and it is well known that the effect can be accumulative in animals under certain conditions. In addition, the dynamic interaction of many factors makes this process even more complex. These factors include environmental conditions, genotype, coat color, health status, degree of acclimatization, nutritional management, cattle handling, and body condition.

## 5. Conclusions

The thermal comfort indices showed a high and positive correlation with TT, especially the THI_PST_ index, although these relationships seem to vary with the location. Thermal indices showed weak relationships with the observed RR, even though they were significant and positive, confirming the different roles that TT and RR have as indicators of heat stress. The THI_PST_ was the unique index that matched well with the increase in the TT observed across the years. In addition, Eigenberg’s equation was a useful tool. However, it underestimated the true RR. Further research that assesses the effects of thermal indices based on pen surface temperature should be conducted in the future.

## Figures and Tables

**Figure 1 animals-13-01169-f001:**
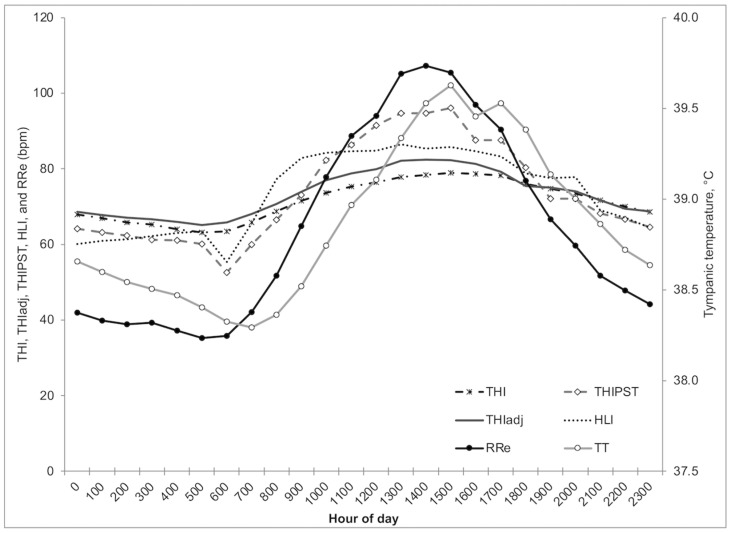
Relationships among tympanic temperatures, respiration rates, and comfort indices for the whole dataset (THI = temperature humidity index, THI_adj_ = adjusted THI, THI_PST_ = THI based on pen surface temperature, HLI = heat load index, RRe = estimated respiration rate, and TT = tympanic temperature).

**Figure 2 animals-13-01169-f002:**
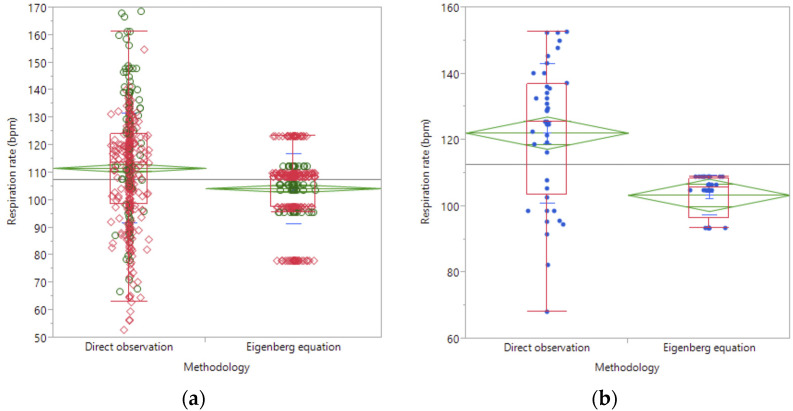
Comparison of the observed and estimated respiration rate (Eigenber’s equation) for the period 1200 to 1500 h in Concord and Socttsbluf (**a**) for the years 2007 (open circles) and 2008 (open diamonds), and only for Scottsbluff, NE (**b**) for 2007 (close dots). In both figures, the top and bottom of the big diamonds are a 95% confidence interval for the mean, whereas the diamonds wide rep-resent the sample average.

**Figure 3 animals-13-01169-f003:**
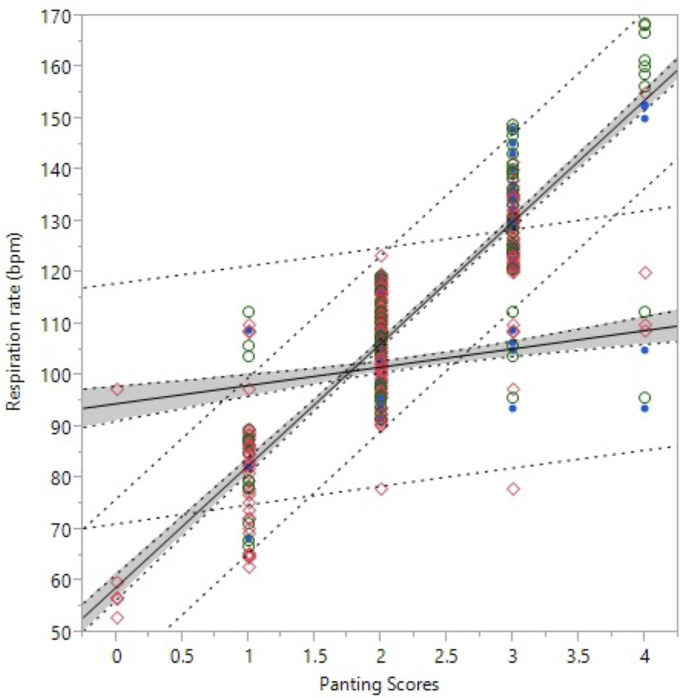
Linear regressions of the observed respiration rates and panting scores for the period 1200 to 1500 h (Respiration rate, bpm = 58.9840 + (23.7504 × Panting score); Adj r^2^ = 0.82, *p* < 0.001) and of estimated respiration rate (Eigenber’s equation), Respiration rate (bpm) = 94.6553 + (3.5663 × Panting score), Adj r^2^ = 0.05, *p* < 0.001).

**Table 1 animals-13-01169-t001:** Panting scores assigned to steers.

Score	Description
0	Normal respiration
1	Elevated respiration
2	Moderate panting and/or presence of drool or small amount of saliva
3	Heavy open-mouthed panting; saliva usually present.
4	Severe open-mouthed panting accompanied by protruding tongue and excessive salivation; usually with neck extended forward

(After Mader [7]).

**Table 2 animals-13-01169-t002:** Summary of mean, SEM, maximum, and minimum values for the main climatic variables and tympanic temperatures for Concord and Scottsbluff ^1^.

Year and Location	RH (%)	WS (m/s)	THI	THI_adj_	THI_PST_	HLI	ST	PST	Ta	BG	TT	SWin	SWout	LWin	LWout
							(°C)					(W/m^−2^)			
2007, Concord															
Mean Max Min SEM ^2^	67.1 94.4 35.1 1.6	2.4 7.3 0.5 0.1	72.0 83.7 52.2 0.6	73.9 88.3 53.3 0.6	77.7 102.3 52.6 1.0	70.5 95.4 45.4 1.2	28.2 32.6 23.1 0.5	28.8 47.8 11.3 0.8	24.5 35.3 11.0 0.5	27.2 42.3 8.8 0.8	39.1 40.5 38.4 0.05	305.3 991.0 −8.4 30.1	56.3 187.0 −1.1 5.4	372.9 446,7 296.3 3.0	448.6 529.2 364.0 3.7
2008, Concord															
Mean Max Min SEM	79.8 100.0 35.8 1.6	2.5 4.5 0.7 0.1	74.1 83.0 61.5 0.6	75.7 86.9 61.5 0.7	70.9 98.4 55.6 1.1	80.1 171.1 52.5 2.4	26.2 28.0 24.3 0.4	23.9 43.1 13.0 0.9	24.8 32.2 16.5 0.4	30.5 80.1 15.4 1.3	38.7 40.4 37.8 0.06	298.8 958.2 −5.7 36.7	50.6 167.4 −0.4 6.2	405.3 456.3 350.7 2.8	483.7 630.1 404.5 6.8
2008, Scottsbluff															
Mean Max Min SEM	66.1 99.6 28 2.2	2.3 7.2 0.5 0.1	70.0 80.4 57.1 0.7	71.8 86.6 59.1 0.8	74.3 120.9 39.8 2.2	72.9 99.8 50.0 1.3	27.4 34.1 21.7 0.6	27.3 61.2 4.2 1.7	23.4 33.4 13.9 0.6	28.4 99.8 10.9 0.9	38.8 40.3 37.2 0.07	281.8 989.8 −4.1 37.1	28.5 132.7 −0.2 3.8	389.8 441.8 336.3 3.1	454.1 629.3 388.0 6.3
Pooled locations															
Mean Max Min SEM	70.3 100.0 28.0 1.1	2.4 7.3 0.5 0.1	72.0 83.7 52.2 0.4	73.8 88.3 53.3 0.4	74.8 120.9 39.8 0.8	73.9 171.1 45.4 1.0	27.4 34.1 21.7 0.2	27.0 61.2 4.2 0.6	24.3 35.3 11.0 0.4	28.5 80.8 8.8 0.5	38.9 40.5 37.2 0.03	296.5 991.0 −8.4 19.7	46.4 187.0 −1.0 3.2	286.9 456.3 296.3 1.9	459.9 630.1 364.0 3.2

^1^ Abbreviations: RH = relative humidity, WS = wind speed, THI = temperature humidity index, THI_adj_ = adjusted THI, THI_PST_ = THI based on pen surface temperature, HLI = heat load index, ST = soil temperature at 10 cm depth, PST = pen surface temperature, Ta = air temperature, BG = black globe temperature, TT = tympanic temperature, SWin = incoming shortwave radiation, SWout = outgoing shortwave radiation, LWin = incoming long-wave radiation, and LWout = outgoing long-wave radiation. ^2^ SEM = standard error of the mean.

**Table 3 animals-13-01169-t003:** Correlation coefficients of tympanic temperature, respiration rate, and comfort indices for the whole dataset.

	TT	THI	THI_adj_	THI_PST_	HLI	RRe
TT	1.00					
THI	0.68	1.00				
THI_adj_	0.70	0.93	1.00			
THI_PST_	0.75	0.77	0.82	1.00		
HLI	0.63	0.51	0.57	0.43	1.00	
RRe	0.68	0.93	0.96	0.85	0.59	1.00

All correlations are significant (*p* < 0.0001). Abbreviations: TT = tympanic temperature, THI = temperature humidity index, THI_adj_ = adjusted THI, THI_PST_ = THI based on pen surface temperature, HLI = heat load index, and RRe = estimated respiration rate based on Eigenberg’s equation.

**Table 4 animals-13-01169-t004:** Correlation coefficients of tympanic temperature, respiration rate, and comfort indices for the summer period in Concord, NE (2007–2008).

	TT	THI	THI_adj_	THI_PST_	HLI	RRe
TT	1.00					
THI	0.68	1.00				
THI_adj_	0.70	0.93	1.00			
THI_PST_	0.80	0.79	0.84	1.00		
HLI	0.22	0.45	0.53	0.42	1.00	
RRe	0.94	0.94	0.95	0.87	0.55	1.00

All correlations are significant (*p* < 0.0001). Abbreviations: TT = tympanic temperature, THI = temperature humidity index, THI_adj_ = adjusted THI, THI_PST_ = THI based on pen surface temperature, HLI = heat load index, and RRe = estimated respiration rate based on Eigenberg’s equation.

**Table 5 animals-13-01169-t005:** Correlation coefficients of tympanic temperature, respiration rate, and comfort indices for the summer period in Scottsbluff, NE (2008).

	TT	THI	THI_adj_	THIPST	HLI	RRe
TT	1.00					
THI	0.63	1.00				
THI_adj_	0.63	0.93	1.00			
THI_PST_	0.70	0.79	0.84	1.00		
HLI	0.52	0.73	0.76	0.61	1.00	
RRe	0.69	0.91	0.97	0.88	0.75	1.00

All correlations are significant (*p* < 0.0001). Abbreviations: TT = tympanic temperature, THI = temperature humidity index, THI_adj_ = adjusted THI, THI_PST_ = THI based on pen surface temperature, HLI = heat load index, and RRe = estimated respiration rate based on Eigenberg’s equation.

## Data Availability

The data that support the study findings are available upon request and after authorization by the authors.

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
