# Peer review of "Evaluation of Four Thermal Comfort Indices and Their Relationship with Physiological Variables in Feedlot Cattle"

_animals, 2023, doi:10.3390/ani13071169_

Round 1
Reviewer 1 Report
I think overall this manuscript was well written. I may possibly consider making it a short communication though. There are some minor grammatical errors and I do have some other minor comments below.
Line 91-94 - How was pen surface temperature collected?
Discussion - I think you need to discuss the limits of your physiological data. Ear temperature is not the best representation of body temperature and only measuring respiration rate at one time during the day limits the ability to assess relationships with this data. I understand it is very labour intensive to collect this data and I feel what you did is still valid I just think you need to add a bit of discussion on it's limits. I think some of the results you saw could be due to this.

Author Response
We thank the editor and the reviewers for all the contributions to our manuscript, which undoubtedly helped to improve its quality. We have responded to all comments and requests from the referees. All changes to the manuscript were made using the Microsoft Word change tracking system. Specific responses to reviewer comments are presented below:
Commentaries Reviewer 1
I think overall this manuscript was well written. I may possibly consider making it a short communication though. There are some minor grammatical errors and I do have some other minor comments below.
Line 91-94 - How was pen surface temperature collected?
R: We added a full description of the equipments used to collect data in both locations (line 110-129)
Discussion - I think you need to discuss the limits of your physiological data. Ear temperature is not the best representation of body temperature and only measuring respiration rate at one time during the day limits the ability to assess relationships with this data. I understand it is very labour intensive to collect this data and I feel what you did is still valid I just think you need to add a bit of discussion on it's limits. I think some of the results you saw could be due to this.
Abstract
L11: change “a” to “have”
R: Done (line 12)
L20: Including text.
R: Done (line 21)
L23: Observation
R: We have included the abbreviation (line 21).
Introduction
L55: Modification
R: We changed “men” to “humans” as requested (line 60).

Reviewer 2 Report
I found your manuscript to be well structured and relevant to future research regarding heat stress impact on feedlot cattle. However, I have some questions / comments. Please see the attachement.

Author Response
We thank the editor and the reviewers for all the contributions to our manuscript, which undoubtedly helped to improve its quality. We have responded to all comments and requests from the referees. All changes to the manuscript were made using the Microsoft Word change tracking system. Specific responses to reviewer comments are presented below:
Commentaries Reviewer 2
- “I assume the abbreviation,, PST“in the Abstract stands for pen surface temperature? Nevertheless, PST should be defined in the Abstract for better understanding. Also, in table 2 the abbreviation for pen surface temperature is ,,SPST “. However, in the first line of the table it sais ,, SST“. The abbreviation of THI estimated using pen surface temperature is (THIPST) (simple summary). However, in the discussion it is THIPST. Abbreviations should be checked for uniformity.”
R: We have included the abbreviation in line 21. In addition, Table 2 was also corrected as well as in the simple summary (line 18).
- In chapter 3.1 the range of soil temperature, pen surface temperature (use of abbreviation?), TT and Ta is listed. However, I cannot find the values in the respective table 2. Except for the abbreviation ,,TT“, the definitions of abbreviations given in the caption consistently follow the order in the headings. Why is the definition of the abbreviation,, TT “always given at last?
R: We have included the abbreviation of PST (line 223). The range values are not presented in Table 2, but they can get by subtracting the respective Maximum and Minimum values of each variable. For example, in 2007 in the Concord location, the RH maximum (94.4%) minus the RH minimum (35.1%) gives us the range of 59.3%. The same applies to the other ranges of values commented. In addition, we have changed the order of the abbreviation following the comments of the reviewer in all the Tables.
- Please change ,, whit“into with (line 145).
R: Done (line 227).
- Please change 0.71 into 0.70 (line 167). The value in the text differs from the value shown in table 4.
R: Done, in the new manuscript, it corresponds to line 260.
- In Figure 2 the indication given for Figure 2a is (a) and for 2b is (b). In the Figure description it sais (A) and (B).
R: Done (line 309)
- The Author Contributions require editing.
R: Done (lines 444-448)

Reviewer 3 Report
In a changing climate, heat stress is an increasing animal health and welfare threat. Conventional measures of heat stress risk (e.g., THI) have limitations. This paper makes a valuable contribution to livestock health and welfare assessment by investigating adjusted measurements to better predict heat stress risk in cattle.
Line 47-59 This introductory paragraph focuses on heat stress as a production issue but it is also a significant animal health and welfare issue. It would be valuable here to include another brief introductory paragraph on heat stress as an animal welfare issue in terms of the Five Domains. It would also be pertinent to highlight there that you have measured both environmental (e.g., SR, WS) & animal-based animal welfare indicators (e.g., RR, panting scores) which is a strength of your study compared to studies that only do one or the other.
Line 70 It would be informative to include some information about cattle’s climatic envelope.
Line 55, line 224 Minor point but change ‘men/man’ to humans
Line 62-65 Important point about the limitations of THI
Line 73 Are you able to include a very brief description of the feedlot facilities (e.g., layout, shade, other cooling systems, materials, stocking density) as this may give readers a better idea of the setup & also understand how translatable your results are to their context
Line 105 Breed?
Line 248 Your analysis indicated that there were more hours of thermal indices above “danger” for THI (> 79) and “extreme caution” for HLI (> 92) during 2007, and this was positively correlated with tympanic temperature. These represent significant animal health & welfare incidents, which under some jurisdictions would warrant mandatory action. Was anything done to address this at the time? Or are you able to include comment as to what you recommend should be done to prevent/during these incidents. That is, after all, the importance of the adjusted tools you’re investigating, yes?
Line 261-263 Good inclusion of possible confounders
Line 278-290 Are you able to speculate on any other factors apart from study design (e.g., animals’ thermoregulatory capacity? Behavioural/physiological differences?) that may have contributed to differences in the correlation coefficients in your compared to previous studies?
Line 298 “These differences reveal the complexity of obtaining a single index that attempts to accurately estimate the risk of heat stress”. This is such an important point, worthy of more discussion. Can you comment on the use of single/over-simplified animal welfare indicators (e.g., just RR), and what your findings show about the limitations of that approach? So often you hear “they’re not breathing up/they look fine, so they are fine” when in reality, as your study shows, that is NOT a valid or comprehensive heat stress risk assessment.
Line 305 Relationships seem to vary with the location. Are you able to include a bit more in the discussion about why you think this might be? And what it means for heat stress risk assessment across different locations?
Author Response
We thank the editor and the reviewers for all the contributions to our manuscript, which undoubtedly helped to improve its quality. We have responded to all comments and requests from the referees. All changes to the manuscript were made using the Microsoft Word change tracking system. Specific responses to reviewer comments are presented below:
Commentaries Reviewer 3
In a changing climate, heat stress is an increasing animal health and welfare threat. Conventional measures of heat stress risk (e.g., THI) have limitations. This paper makes a valuable contribution to livestock health and welfare assessment by investigating adjusted measurements to better predict heat stress risk in cattle.
Line 47-59 This introductory paragraph focuses on heat stress as a production issue but it is also a significant animal health and welfare issue. It would be valuable here to include another brief introductory paragraph on heat stress as an animal welfare issue in terms of the Five Domains. It would also be pertinent to highlight there that you have measured both environmental (e.g., SR, WS) & animal-based animal welfare indicators (e.g., RR, panting scores) which is a strength of your study compared to studies that only do one or the other.
R: Done (lines 66 to 78)
Line 70 It would be informative to include some information about cattle’s climatic envelope.
R: Could the referee be more specific? Is she/he asking for a description of the weather conditions?
Line 55, line 224 Minor point but change ‘men/man’ to humans
R: Done (line 61).
Line 62-65 Important point about the limitations of THI
R: No comments.
Line 73 Are you able to include a very brief description of the feedlot facilities (e.g., layout, shade, other cooling systems, materials, stocking density) as this may give readers a better idea of the setup & also understand how translatable your results are to their context.
R: We have included a new paragraph with a brief description of the facilities (lines 92-101)
Line 105 Breed?
R: The breed was added (lines 171-172)
Line 248 Your analysis indicated that there were more hours of thermal indices above “danger” for THI (> 79) and “extreme caution” for HLI (> 92) during 2007, and this was positively correlated with tympanic temperature. These represent significant animal health & welfare incidents, which under some jurisdictions would warrant mandatory action. Was anything done to address this at the time? Or are you able to include comments as to what you recommend should be done to prevent/during these incidents? That is, after all, the importance of the adjusted tools you’re investigating, yes?
R: No actions were taken under those conditions at that moment. Nevertheless, a sentence was added (lines 357-360).
Line 261-263 Good inclusion of possible confounders
R: No comments.
Line 278-290 Are you able to speculate on any other factors apart from study design (e.g., animals’ thermoregulatory capacity? Behavioural/physiological differences?) that may have contributed to differences in the correlation coefficients in your compared to previous studies?
R: A paragraph was added (lines 399-410).
Line 298 “These differences reveal the complexity of obtaining a single index that attempts to accurately estimate the risk of heat stress”. This is such an important point, worthy of more discussion. Can you comment on the use of single/over-simplified animal welfare indicators (e.g., just RR), and what your findings show about the limitations of that approach? So often you hear “they’re not breathing up/they look fine, so they are fine” when in reality, as your study shows, that is NOT a valid or comprehensive heat stress risk assessment.
R: A paragraph was added (lines 419-428).
Line 305 Relationships seem to vary with the location. Are you able to include a bit more in the discussion about why you think this might be? And what it means for heat stress risk assessment across different locations?
R: Done (lines 399-410)
